# Graphical Models for Inference with Missing Data

**Karthika Mohan**
Dept. of Computer Science
Univ. of California, Los Angeles
Los Angeles, CA 90095
karthika@cs.ucla.edu

**Judea Pearl**
Dept. of Computer Science
Univ. of California, Los Angeles
Los Angeles, CA 90095
judea@cs.ucla.edu

**Jin Tian**
Dept. of Computer Science
Iowa State University
Ames, IA 50011
jtian@iastate.edu

## Abstract

We address the problem of *recoverability* i.e. deciding whether there exists a consistent estimator of a given relation $Q$, when data are missing not at random. We employ a formal representation called 'Missingness Graphs' to explicitly portray the causal mechanisms responsible for missingness and to encode dependencies between these mechanisms and the variables being measured. Using this representation, we derive conditions that the graph should satisfy to ensure recoverability and devise algorithms to detect the presence of these conditions in the graph.

## 1 Introduction

The "missing data" problem arises when values for one or more variables are missing from recorded observations. The extent of the problem is evidenced from the vast literature on missing data in such diverse fields as social science, epidemiology, statistics, biology and computer science. Missing data could be caused by varied factors such as high cost involved in measuring variables, failure of sensors, reluctance of respondents in answering certain questions or an ill-designed questionnaire. Missing data also plays a major role in survival data analysis and has been treated primarily using Kaplan-Meier estimation [30].

In machine learning, a typical example is the Recommender System [16] that automatically generates a list of products that are of potential interest to a given user from an incomplete dataset of user ratings. Online portals such as Amazon and eBay employ such systems. Other areas such as data mining [7], knowledge discovery [18] and network tomography [2] are also plagued by missing data problems. Missing data can have several harmful consequences [23, 26]. Firstly they can significantly bias the outcome of research studies. This is mainly because the response profiles of non-respondents and respondents can be significantly different from each other. Hence ignoring the former distorts the true proportion in the population. Secondly, performing the analysis using only complete cases and ignoring the cases with missing values can reduce the sample size thereby substantially reducing estimation efficiency. Finally, many of the algorithms and statistical techniques are generally tailored to draw inferences from complete datasets. It may be difficult or even inappropriate to apply these algorithms and statistical techniques on incomplete datasets.

### 1.1 Existing Methods for Handling Missing Data

There are several methods for handling missing data, described in a rich literature of books, articles and software packages, which are briefly summarized here[1]. Of these, listwise deletion and pairwise deletion are used in approximately $96\%$ of studies in the social and behavioral sciences [24].

Listwise deletion refers to a simple method in which cases with missing values are deleted [3]. Unless data are missing completely at random, listwise deletion can bias the outcome [31]. Pairwise

deletion (or "available case") is a deletion method used for estimating pairwise relations among variables. For example, to compute the covariance of variables $X$ and $Y$, all those cases or observations in which both $X$ and $Y$ are observed are used, regardless of whether other variables in the dataset have missing values.

The expectation-maximization (EM) algorithm is a general technique for finding maximum likelihood (ML) estimates from incomplete data. It has been proven that likelihood-based inference while ignoring the missing data mechanism, leads to unbiased estimates under the assumption of *missing at random (MAR)* [13]. Most work in machine learning assumes MAR and proceeds with ML or Bayesian inference. Exceptions are recent works on collaborative filtering and recommender systems which develop probabilistic models that explicitly incorporate missing data mechanism [16, 14, 15]. ML is often used in conjunction with imputation methods, which in layman terms, substitutes a reasonable guess for each missing value [1]. A simple example is *Mean Substitution*, in which all missing observations of variable $X$ are substituted with the mean of all observed values of $X$. Hot-deck imputation, cold-deck imputation [17] and Multiple Imputation [26, 27] are examples of popular imputation procedures. Although these techniques work well in practice, performance guarantees (eg: convergence and unbiasedness) are based primarily on simulation experiments.

Missing data discussed so far is a special case of *coarse data*, namely data that contains observations made in the power set rather than the sample space of variables of interest [12]. The notion of coarsening at random (CAR) was introduced in [12] and identifies the condition under which coarsening mechanism can be ignored while drawing inferences on the distribution of variables of interest [10]. The notion of sequential CAR has been discussed in [9]. For a detailed discussion on coarsened data refer to [30].

Missing data literature leaves many unanswered questions with regard to theoretical guarantees for the resulting estimates, the nature of the assumptions that must be made prior to employing various procedures and whether the assumptions are testable. For a gentle introduction to the missing data problem and the issue of testability refer to [22, 19]. This paper aims to illuminate missing data problems using causal graphs [See Appendix 5.2 for justification]. The questions we pose are: Given a target relation $Q$ to be estimated and a set of assumptions about the missingness process encoded in a graphical model, under what conditions does a consistent estimate exist and how can we elicit it from the data available?

We answer these questions with the aid of Missingness Graphs ($m$-graphs in short) to be described in Section 2. Furthermore, we review the traditional taxonomy of missing data problems and cast it in graphical terms. In Section 3 we define the notion of recoverability - the existence of a consistent estimate - and present graphical conditions for detecting recoverability of a given probabilistic query $Q$. Conclusions are drawn in Section 4.

## 2 Graphical Representation of the Missingness Process

### 2.1 Missingness Graphs

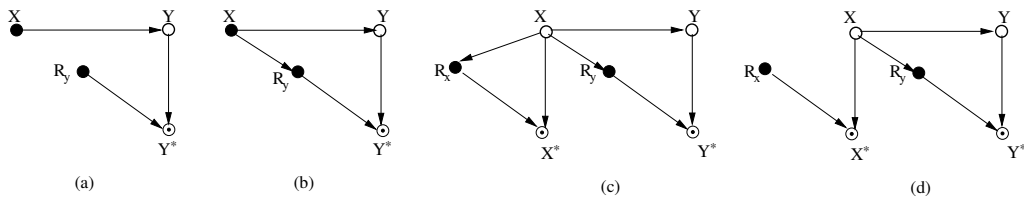

Figure 1: $m$-graphs for data that are: (a) MCAR, (b) MAR, (c) & (d) MNAR; Hollow and solid circles denote partially and fully observed variables respectively.

Graphical models such as DAGs (Directed Acyclic Graphs) can be used for encoding as well as portraying conditional independencies and causal relations, and the graphical criterion called $d$-separation (refer Appendix-5.1, Definition-3) can be used to read them off the graph [21, 20]. Graphical Models have been used to analyze missing information in the form of missing cases (due to sample selection bias)[4]. Using causal graphs, [8]- analyzes missingness due to attrition (partially

observed outcome) and [29]- cautions against the indiscriminate use of auxiliary variables. In both papers missing values are associated with one variable and interactions among several missingness mechanisms remain unexplored.

The need exists for a general approach capable of modeling an arbitrary data-generating process and deciding whether (and how) missingness can be outmaneuvered in every dataset generated by that process. Such a general approach should allow each variable to be governed by its own missingness mechanism, and each mechanism to be triggered by other (potentially) partially observed variables in the model. To achieve this flexibility we use a graphical model called "missingness graph" ($m$-graph, for short) which is a DAG (Directed Acyclic Graph) defined as follows.

Let $G(\mathbb{V}, E)$ be the causal DAG where $\mathbb{V} = V \cup U \cup V^* \cup \mathbb{R}$. $V$ is the set of observable nodes. Nodes in the graph correspond to variables in the data set. $U$ is the set of unobserved nodes (also called latent variables). $E$ is the set of edges in the DAG. Oftentimes we use bi-directed edges as a shorthand notation to denote the existence of a $U$ variable as common parent of two variables in $V_o \cup V_m \cup \mathbb{R}$. $V$ is partitioned into $V_o$ and $V_m$ such that $V_o \subseteq V$ is the set of variables that are observed in all records in the population and $V_m \subseteq V$ is the set of variables that are missing in at least one record. Variable $X$ is termed as *fully observed* if $X \in V_o$ and *partially observed* if $X \in V_m$.

Associated with every partially observed variable $V_i \in V_m$ are two other variables $R_{v_i}$ and $V_i^*$, where $V_i^*$ is a proxy variable that is actually observed, and $R_{v_i}$ represents the status of the causal mechanism responsible for the missingness of $V_i^*$; formally,

$$v_i^* = f(r_{v_i}, v_i) = \begin{cases} v_i & \text{if } r_{v_i} = 0 \\ m & \text{if } r_{v_i} = 1 \end{cases} \tag{1}$$

Contrary to conventional use, $R_{v_i}$ is not treated merely as the missingness *indicator* but as a driver (or a switch) that enforces equality between $V_i$ and $V_i^*$. $V^*$ is a set of all proxy variables and $\mathbb{R}$ is the set of all causal mechanisms that are responsible for missingness. $R$ variables may not be parents of variables in $V \cup U$. This graphical representation succinctly depicts both the causal relationships among variables in $V$ and the process that accounts for missingness in some of the variables. We call this graphical representation **Missingness Graph** or $m$-graph for short. Since every d-separation in the graph implies conditional independence in the distribution [21], the m-graph provides an effective way of representing the statistical properties of the missingness process and, hence, the potential of recovering the statistics of variables in $V_m$ from partially missing data.

## 2.2 Taxonomy of Missingness Mechanisms

It is common to classify missing data mechanisms into three types [25, 13]:
**Missing Completely At Random (MCAR) :** Data are MCAR if the probability that $V_m$ is missing is independent of $V_m$ or any other variable in the study, as would be the case when respondents decide to reveal their income levels based on coin-flips.
**Missing At Random (MAR) :** Data are MAR if for all data cases $Y$, $P(R|Y_{obs}, Y_{mis}) = P(R|Y_{obs})$ where $Y_{obs}$ denotes the observed component of $Y$ and $Y_{mis}$, the missing component. Example: Women in the population are more likely to not reveal their age.
**Missing Not At Random (MNAR)** or "non-ignorable missing": Data that are neither MAR nor MCAR are termed as MNAR. Example: Online shoppers rate an item with a high probability either if they love the item or if they loathe it. In other words, the probability that a shopper supplies a rating is dependent on the shopper's underlying liking [16].

Because it invokes specific values of the observed and unobserved variables, (i.e., $Y_{obs}$ and $Y_{mis}$), many authors find Rubin's definition difficult to apply in practice and prefer to work with definitions expressed in terms of independencies among variables (see [28, 11, 6, 17]). In the *graph-based* interpretation used in this paper, MCAR is defined as total independence between $\mathbb{R}$ and $V_o \cup V_m \cup U$ i.e. $\mathbb{R} \perp\!\!\!\perp (V_o \cup V_m \cup U)$, as depicted in Figure 1(a). MAR is defined as independence between $\mathbb{R}$ and $V_m \cup U$ given $V_o$ i.e. $\mathbb{R} \perp\!\!\!\perp V_m \cup U | V_o$, as depicted in Figure 1(b). Finally if neither of these conditions hold, data are termed MNAR, as depicted in Figure 1(c) and (d). This graph-based interpretation uses slightly stronger assumptions than Rubin's, with the advantage that the user can comprehend, encode and communicate the assumptions that determine the classification of the problem. Additionally, the conditional independencies that define each class are represented explicitly as separation conditions

in the corresponding m-graphs. We will use this taxonomy in the rest of the paper, and will label data MCAR, MAR and MNAR according to whether the defining conditions, $\mathbb{R} \perp\!\!\!\perp V_o \cup V_m \cup U$ (for MCAR), $\mathbb{R} \perp\!\!\!\perp V_m \cup U | V_o$ (for MAR) are satisfied in the corresponding m-graphs.

## 3  Recoverability

In this section we will examine the conditions under which a bias-free estimate of a given probabilistic relation $Q$ can be computed. We shall begin by defining the notion of recoverability.

**Definition 1** (Recoverability). *Given a $m$-graph $G$, and a target relation $Q$ defined on the variables in $V$, $Q$ is said to be recoverable in $G$ if there exists an algorithm that produces a consistent estimate of $Q$ for every dataset $D$ such that $P(D)$ is (1) compatible with $G$ and (2) strictly positive over complete cases i.e. $P(V_o, V_m, \mathbb{R} = 0) > 0$.[2]*

Here we assume that the observed distribution over complete cases $P(V_o, V_m, R = 0)$ is strictly positive, thereby rendering recoverability a property that can be ascertained exclusively from the m-graph.

**Corollary 1.** *A relation $Q$ is recoverable in $G$ if and only if $Q$ can be expressed in terms of the probability $P(O)$ where $O = \{R, V^*, V_o\}$ is the set of observable variables in $G$. In other words, for any two models $M_1$ and $M_2$ inducing distributions $P^{M_1}$ and $P^{M_2}$ respectively, if $P^{M_1}(O) = P^{M_2}(O) > 0$ then $Q^{M_1} = Q^{M_2}$.*

Proof: (sketch) The corollary merely rephrases the requirement of obtaining a consistent estimate to that of expressibility in terms of observables.

Practically, what recoverability means is that if the data $D$ are generated by any process compatible with $G$, a procedure exists that computes an estimator $\hat{Q}(D)$ such that, in the limit of large samples, $\hat{Q}(D)$ converges to $Q$. Such a procedure is called a "consistent estimator." Thus, recoverability is the sole property of $G$ and $Q$, not of the data available, or of any routine chosen to analyze or process the data.

**Recoverability when data are MCAR**  For MCAR data we have $\mathbb{R} \perp\!\!\!\perp (V_o \cup V_m)$. Therefore, we can write $P(V) = P(V|\mathbb{R}) = P(V_o, V^*|\mathbb{R} = 0)$. Since both $\mathbb{R}$ and $V^*$ are observables, the joint probability $P(V)$ is consistently estimable (hence recoverable) by considering complete cases only (listwise deletion), as shown in the following example.

**Example 1.** *Let $X$ be the treatment and $Y$ be the outcome as depicted in the $m$-graph in Fig. 1 (a). Let it be the case that we accidentally deleted the values of $Y$ for a handful of samples, hence $Y \in V_m$. Can we recover $P(X, Y)$?*
*From $D$, we can compute $P(X, Y^*, R_y)$. From the $m$-graph $G$, we know that $Y^*$ is a collider and hence by d-separation, $(X \cup Y) \perp\!\!\!\perp R_y$. Thus $P(X, Y) = P(X, Y|R_y)$. In particular, $P(X, Y) = P(X, Y|R_y = 0)$. When $R_y = 0$, by eq. (1), $Y^* = Y$. Hence,*

$$P(X, Y) = P(X, Y^*|R_y = 0) \tag{2}$$

*The RHS of Eq. 2 is consistently estimable from $D$; hence $P(X, Y)$ is recoverable.*

**Recoverability when data are MAR**  When data are MAR, we have $\mathbb{R} \perp\!\!\!\perp V_m | V_o$. Therefore $P(V) = P(V_m|V_o)P(V_o) = P(V_m|V_o, R = 0)P(V_o)$. Hence the joint distribution $P(V)$ is recoverable.

**Example 2.** *Let $X$ be the treatment and $Y$ be the outcome as depicted in the $m$-graph in Fig. 1 (b). Let it be the case that some patients who underwent treatment are not likely to report the outcome, hence the arrow $X \rightarrow R_y$. Under the circumstances, can we recover $P(X, Y)$?*

*From $D$, we can compute $P(X, Y^*, R_y)$. From the $m$-graph $G$, we see that $Y^*$ is a collider and $X$ is a fork. Hence by d-separation, $Y \perp\!\!\!\perp R_y | X$. Thus $P(X, Y) = P(Y|X)P(X) = P(Y|X, R_y)P(X)$.*

*In particular, $P(X,Y) = P(Y|X, R_y = 0)P(X)$. When $R_y = 0$, by eq. (1), $Y^* = Y$. Hence,*

$$P(X,Y) = P(Y^*|X, R_y = 0)P(X) \qquad (3)$$

*and since $X$ is fully observable, $P(X,Y)$ is recoverable.*

Note that eq. (2) permits $P(X,Y)$ to be recovered by listwise deletion, while eq. (3) does not; it requires that $P(X)$ be estimated *first* over all samples, including those in which $Y$ is missing. In this paper we focus on recoverability under large sample assumption and will not be dealing with the shrinking sample size issue.

**Recoverability when data are MNAR**    Data that are neither MAR nor MCAR are termed MNAR. Though it is generally believed that relations in MNAR datasets are not recoverable, the following example demonstrates otherwise.

**Example 3.** *Fig. 1 (d) depicts a study where (i) some units who underwent treatment ($X = 1$) did not report the outcome ($Y$) and (ii) we accidentally deleted the values of treatment for a handful of cases. Thus we have missing values for both $X$ and $Y$ which renders the dataset MNAR. We shall show that $P(X,Y)$ is recoverable. From D, we can compute $P(X^*, Y^*, R_x, R_y)$. From the m-graph G, we see that $X \perp\!\!\!\perp R_x$ and $Y \perp\!\!\!\perp (R_x \cup R_y)|X$. Thus $P(X,Y) = P(Y|X)P(X) = P(Y|X, R_y = 0, R_x = 0)P(X|R_x = 0)$. When $R_y = 0$ and $R_x = 0$ we have (by Equation (1) ), $Y^* = Y$ and $X^* = X$. Hence,*

$$P(X,Y) = P(Y^*|X^*, R_x = 0, R_y = 0)P(X^*|R_x = 0) \qquad (4)$$

*Therefore, $P(X,Y)$ is recoverable.*

The estimand in eq. (4) also dictates how $P(X,Y)$ should be estimated from the dataset. In the first step, we delete all cases in which $X$ is missing and create a new data set $D'$ from which we estimate $P(X)$. Dataset $D'$ is further pruned to form dataset $D''$ by removing all cases in which $Y$ is missing. $P(Y|X)$ is then computed from $D''$. Note that **order matters**; had we deleted cases in the reverse order, $Y$ and then $X$, the resulting estimate would be biased because the $d$-separations needed for establishing the validity of the estimand: $P(X|Y)P(Y)$, are not supported by $G$. We will call this sequence of deletions as *deletion order*.

Several features are worth noting regarding this *graph-based* taxonomy of missingness mechanisms. First, although MCAR and MAR can be verified by inspecting the $m$-graph, they cannot, in general be verified from the data alone. Second, the assumption of MCAR allows an estimation procedure that amounts (asymptotically) to listwise deletion, while MAR dictates a procedure that amounts to listwise deletion in every stratum of $V_o$. Applying MAR procedure to MCAR problem is safe, because all conditional independencies required for recoverability under the MAR assumption also hold in an MCAR problem, i.e. $\mathbb{R} \perp\!\!\!\perp (V_o, V_m) \Rightarrow \mathbb{R} \perp\!\!\!\perp V_m|V_o$. The converse, however, does not hold, as can be seen in Fig. 1 (b). Applying listwise deletion is likely to result in bias, because the necessary condition $\mathbb{R} \perp\!\!\!\perp (V_o, V_m)$ is violated in the graph. An interesting property which evolves from this discussion is that recoverability of certain relations does not require $R_{V_i} \perp\!\!\!\perp V_i|V_o$ ; a subset of $V_o$ would suffice as shown below.

**Property 1.**  $P(V_i)$ is recoverable if $\exists W \subseteq V_o$ such that $R_{V_i} \perp\!\!\!\perp V|W$.

*Proof:*  $P(V_i)$ may be decomposed as: $P(V_i) = \sum_w P(V_i^*|R_{v_i} = 0, W)P(W)$ since $V_i \perp\!\!\!\perp R_{V_i}|W$ and $W \subseteq V_o$. Hence $P(V_i)$ is recoverable.

It is important to note that the recoverability of $P(X,Y)$ in Fig. 1(d) was feasible despite the fact that the missingness model would not be considered Rubin's MAR (as defined in [25]). In fact, an overwhelming majority of the data generated by each one of our MNAR examples would be outside Rubin's MAR. For a brief discussion on these lines, refer to Appendix- 5.4.

Our next question is: how can we determine if a given relation is recoverable? The following theorem provides a sufficient condition for recoverability.

### 3.1   Conditions for Recoverability

**Theorem 1.** *A query $Q$ defined over variables in $V_o \cup V_m$ is recoverable if it is decomposable into terms of the form $Q_j = P(S_j|T_j)$ such that $T_j$ contains the missingness mechanism $R_v = 0$ of every partially observed variable $V$ that appears in $Q_j$.*

*Proof:* If such a decomposition exists, every $Q_j$ is estimable from the data, hence the entire expression for $Q$ is recoverable.

**Example 4.** *Equation (4) demonstrates a decomposition of $Q = P(X,Y)$ into a product of two terms $Q_1 = P(Y|X, R_x = 0, R_y = 0)$ and $Q_2 = P(X|R_x = 0)$ that satisfy the condition of Theorem 1. Hence $Q$ is recoverable.*

**Example 5.** *Consider the problem of recovering $Q = P(X,Y)$ from the $m$-graph of Fig. 3(b). Attempts to decompose $Q$ by the chain rule, as was done in Eqs. (3) and (4) would not satisfy the conditions of Theorem 1. To witness we write $P(X,Y) = P(Y|X)P(X)$ and note that the graph does not permit us to augment any of the two terms with the necessary $R_x$ or $R_y$ terms; $X$ is independent of $R_x$ only if we condition on $Y$, which is partially observed, and $Y$ is independent of $R_y$ only if we condition on $X$ which is also partially observed. This deadlock can be disentangled however using a non-conventional decomposition:*

$$Q = P(X,Y) = P(X,Y)\frac{P(R_x, R_y|X,Y)}{P(R_x, R_y|X,Y)}$$
$$= \frac{P(R_x, R_y)P(X,Y|R_x, R_y)}{P(R_x|Y, R_y)P(R_y|X, R_x)} \quad (5)$$

*where the denominator was obtained using the independencies $R_x \perp\!\!\!\perp (X, R_y)|Y$ and $R_y \perp\!\!\!\perp (Y, R_x)|X$ shown in the graph. The final expression above satisfies Theorem 1 and renders $P(X,Y)$ recoverable. This example again shows that recovery is feasible even when data are MNAR.*

Theorem 2 operationalizes the decomposability requirement of Theorem 1.

**Theorem 2** (Recoverability of the Joint $P(V)$)**.** *Given a $m$-graph $G$ with no edges between the $R$ variables and no latent variables as parents of $R$ variables, a necessary and sufficient condition for recovering the joint distribution $P(V)$ is that no variable $X$ be a parent of its missingness mechanism $R_X$. Moreover, when recoverable, $P(V)$ is given by*

$$P(v) = \frac{P(R = 0, v)}{\prod_i P(R_i = 0|pa^o_{r_i}, pa^m_{r_i}, R_{Pa^m_{r_i}} = 0)}, \quad (6)$$

*where $Pa^o_{r_i} \subseteq V_o$ and $Pa^m_{r_i} \subseteq V_m$ are the parents of $R_i$.*

*Proof.* (sufficiency) The observed joint distribution may be decomposed according to $G$ as

$$P(R = 0, v) = \sum_u P(v, u)P(R = 0|v, u)$$
$$= P(v) \prod_i P(R_i = 0|pa^o_{r_i}, pa^m_{r_i}), \quad (7)$$

where we have used the facts that there are no edges between the $R$ variables, and that there are no latent variables as parents of $R$ variables. If $V_i$ is not a parent of $R_i$ (i.e. $V_i \notin Pa^m_{r_i}$), then we have $R_i \perp\!\!\!\perp R_{Pa^m_{r_i}}|(Pa^o_{r_i} \cup Pa^m_{r_i})$. Therefore,

$$P(R_i = 0|pa^o_{r_i}, pa^m_{r_i}) = P(R_i = 0|pa^o_{r_i}, pa^m_{r_i}, R_{Pa^m_{r_i}} = 0). \quad (8)$$

Given strictly positive $P(R = 0, V_m, V_o)$, we have that all probabilities $P(R_i = 0|pa^o_{r_i}, pa^m_{r_i}, R_{Pa^m_{r_i}} = 0)$ are strictly positive. Using Equations (7) and (8), we conclude that $P(V)$ is recoverable as given by Eq. (6).

(necessity) If $X$ is a parent of its missingness mechanism $R_X$, then $P(X)$ is not recoverable based on Lemmas 3 and 4 in Appendix 5.5. Therefore the joint $P(V)$ is not recoverable. $\square$

The following theorem gives a sufficient condition for recovering the joint distribution in a Markovian model.

**Theorem 3.** *Given a $m$-graph with no latent variables (i.e., Markovian) the joint distribution $P(V)$ is recoverable if no missingness mechanism $R_X$ is a descendant of its corresponding variable $X$. Moreover, if recoverable, then $P(V)$ is given by*

$$P(v) = \prod_{i, V_i \in V_o} P(v_i|pa^o_i, pa^m_i, R_{Pa^m_i} = 0) \prod_{j, V_j \in V_m} P(v_j|pa^o_j, pa^m_j, R_{V_j} = 0, R_{Pa^m_j} = 0), \quad (9)$$

*where $Pa^o_i \subseteq V_o$ and $Pa^m_i \subseteq V_m$ are the parents of $V_i$.*

Proof: Refer Appendix-5.6

**Definition 2** (Ordered factorization). *An ordered factorization over a set $O$ of ordered $V$ variables $Y_1 < Y_2 < \ldots < Y_k$, denoted by $f(O)$, is a product of conditional probabilities $f(O) = \prod_i P(Y_i|X_i)$ where $X_i \subseteq \{Y_{i+1}, \ldots, Y_n\}$ is a minimal set such that $Y_i \perp\!\!\!\perp (\{Y_{i+1}, \ldots, Y_n\} \setminus X_i)|X_i$.*

**Theorem 4.** *A sufficient condition for recoverability of a relation $Q$ is that $Q$ be decomposable into an ordered factorization, or a sum of such factorizations, such that every factor $Q_i = P(Y_i|X_i)$ satisfies $Y_i \perp\!\!\!\perp (R_{y_i}, R_{x_i})|X_i$. A factorization that satisfies this condition will be called admissible.*

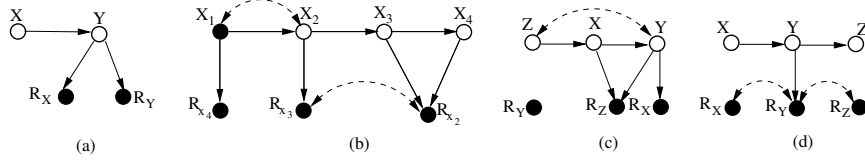

(a)  (b)  (c)  (d)

Figure 2: Graph in which (a) only $P(X|Y)$ is recoverable (b) $P(X_4)$ is recoverable only when conditioned on $X_1$ as shown in Example 6 (c) $P(X, Y, Z)$ is recoverable (d) $P(X, Z)$ is recoverable.

*Proof.* follows from Theorem-1 noting that ordered factorization is one specific form of decomposition. □

Theorem 4 will allow us to confirm recoverability of certain queries $Q$ in models such as those in Fig. 2(a), (b) and (d), which do not satisfy the requirement in Theorem 2. For example, by applying Theorem 4 we can conclude that, (1) in Figure 2 (a), $P(X|Y) = P(X|R_x = 0, R_y = 0, Y)$ is recoverable, (2) in Figure 2 (c), $P(X, Y, Z) = P(Z|X, Y, R_z = 0, R_x = 0, R_y = 0)P(X|Y, R_x = 0, R_y = 0)P(Y|R_y = 0)$ is recoverable and (3) in Figure 2 (d), $P(X, Z) = P(X, Z|Rx = 0, R_z = 0)$ is recoverable.

Note that the condition of Theorem 4 differs from that of Theorem 1 in two ways. Firstly, the decomposition is limited to ordered factorizations i.e. $Y_i$ is a singleton and $X_i$ a set. Secondly, both $Y_i$ and $X_i$ are taken from $V_o \cup V_m$, thus excluding $R$ variables.

**Example 6.** *Consider the query $Q = P(X_4)$ in Fig. 2(b). $Q$ can be decomposed in a variety of ways, among them being the factorizations:*
*(a) $P(X_4) = \sum_{x_3} P(X_4|X_3)P(X_3)$ for the order $X_4, X_3$*
*(b) $P(X_4) = \sum_{x_2} P(X_4|X_2)P(X_2)$ for the order $X_4, X_2$*
*(c) $P(X_4) = \sum_{x_1} P(X_4|X_1)P(X_1)$ for the order $X_4, X_1$*
*Although each of $X_1, X_2$ and $X_3$ d-separate $X_4$ from $R_{X_4}$, only (c) is admissible since each factor satisfies Theorem 4. Specifically, (c) can be written as $\sum_{x_1} P(X_4^*|X_1, R_{X_4} = 0)P(X_1)$ and can be estimated by the deletion schedule $(X_1, X_4)$, i.e., in each stratum of $X_1$, we delete samples for which $R_{X_4} = 1$ and compute $P(X_4^*, R_{x_4} = 0, X_1)$. In (a) and (b) however, Theorem-4 is not satisfied since the graph does not permit us to rewrite $P(X_3)$ as $P(X_3|R_{x_3} = 0)$ or $P(X_2)$ as $P(X_2|R_{x_2} = 0)$.*

### 3.2 Heuristics for Finding Admissible Factorization

Consider the task of estimating $Q = P(X)$, where $X$ is a set, by searching for an admissible factorization of $P(X)$ (one that satisfies Theorem 4), possibly by resorting to additional variables, $Z$, residing outside of $X$ that serve as separating sets. Since there are exponentially large number of ordered factorizations, it would be helpful to rule out classes of non-admissible ordering prior to their enumeration whenever non-admissibility can be detected in the graph. In this section, we provide lemmata that would aid in pruning process by harnessing information from the graph.

**Lemma 1.** *An ordered set $O$ will not yield an admissible decomposition if there exists a partially observed variable $V_i$ in the order $O$ which is not marginally independent of $R_{V_i}$ such that all minimal separators (refer Appendix-5.1, Definition-4) of $V_i$ that d-separate it from $R_{v_i}$ appear before $V_i$.*

Proof: Refer Appendix-5.7

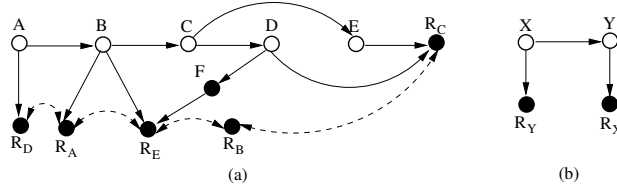

Figure 3: demonstrates (a) pruning in Example-7 (b) $P(X, Y)$ is recoverable in Example-5

Applying lemma-1 requires a solution to a set of disjunctive constraints which can be represented by directed constraint graphs [5].

**Example 7.** *Let $Q = P(X)$ be the relation to be recovered from the graph in Fig. 3 (a). Let $X = \{A, B, C, D, E\}$ and $Z = F$. The total number of ordered factorizations is $6! = 720$. The independencies implied by minimal separators (as required by Lemma-1) are: $A \perp\!\!\!\perp R_A | B$, $B \perp\!\!\!\perp R_B | \phi$, $C \perp\!\!\!\perp R_C | \{D, E\}$, ($D \perp\!\!\!\perp R_D | A$ or $D \perp\!\!\!\perp R_D | C$ or $D \perp\!\!\!\perp R_D | B$) and ($E \perp\!\!\!\perp R_E | \{B, F\}$ or $E \perp\!\!\!\perp R_E | \{B, D\}$ or $E \perp\!\!\!\perp R_E | C$). To test whether (B,A,D,E,C,F) is potentially admissible we need not explicate all 6 variables; this order can be ruled out as soon as we note that A appears after B. Since B is the only minimal separator that d-separates A from $R_A$ and B precedes A, Lemma-1 is violated. Orders such as $(C, D, E, A, B, F)$, $(C, D, A, E, B, F)$ and $(C, E, D, A, F, B)$ satisfy the condition stated in Lemma 1 and are potential candidates for admissibility.*

The following lemma presents a simple test to determine non-admissibility by specifying the condition under which a given order can be summarily removed from the set of candidate orders that are likely to yield admissible factorizations.

**Lemma 2.** *An ordered set $O$ will not yield an admissible decomposition if it contains a partially observed variable $V_i$ for which there exists no set $S \subseteq V$ that d-separates $V_i$ from $R_{V_i}$.*

*Proof:* The factor $P(V_i | V_{i+1}, \ldots, V_n)$ corresponding to $V_i$ can never satisfy the condition required by Theorem 4.

An interesting consequence of Lemma 2 is the following corollary that gives a sufficient condition under which no ordered factorization can be labeled admissible.

**Corollary 2.** *For any disjoint sets $X$ and $Y$, there exists no admissible factorization for recovering the relation $P(Y|X)$ by Theorem 4 if $Y$ contains a partially observed variable $V_i$ for which there exists no set $S \subseteq V$ that d-separates $V_i$ from $R_{V_i}$.*

## 4 Conclusions

We have demonstrated that causal graphical models depicting the data generating process can serve as a powerful tool for analyzing missing data problems and determining (1) if theoretical impediments exist to eliminating bias due to data missingness, (2) whether a given procedure produces consistent estimates, and (3) whether such a procedure can be found algorithmically. We formalized the notion of *recoverability* and showed that relations are always recoverable when data are missing at random (MCAR or MAR) and, more importantly, that in many commonly occurring problems, recoverability can be achieved even when data are missing not at random (MNAR). We further presented a sufficient condition to ensure recoverability of a given relation $Q$ (Theorem 1) and operationalized Theorem 1 using graphical criteria (Theorems 2, 3 and 4). In summary, we demonstrated some of the insights and capabilities that can be gained by exploiting causal knowledge in missing data problems.

## Acknowledgment

This research was supported in parts by grants from NSF #IIS-1249822 and #IIS-1302448 and ONR #N00014-13-1-0153 and #N00014-10-1-0933

## Footnotes

[1]For detailed discussions we direct the reader to the books- [1, 6, 13, 17].

[2]In many applications such as truncation by death, the problem forbids certain combinations of events from occurring, in which case the definition need be modified to accommodate such constraints as shown in Appendix-5.3. Though this modification complicates the definition of "recoverability", it does not change the basic results derived in this paper.

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
