[Supplementary Material]

# 5 Appendix

## 5.1 $d$-separation

**Definition 3** ($d$-separation[3]). *A path $p$ is said to be $d$-separated by a set of nodes $Z$ if and only if:*
*(1) $p$ contains a chain $i \rightarrow m \rightarrow j$ or a fork $i \leftarrow m \rightarrow j$ such that the middle node $m$ is in $Z$, or*
*(2) $p$ contains an inverted fork (or collider) $i \rightarrow m \leftarrow j$ such that the middle node $m$ is not in $Z$ and such that no descendant of $m$ is in $Z$.*

*A set $Z$ is said to $d$-separate $X$ from $Y$ if and only if $Z$ blocks every path from a node in $X$ to a node in $Y$ and is denoted by $Y \perp\!\!\!\perp X | Z$.*

**Definition 4** (Minimal Separator). *Given two sets of nodes $X$ and $Y$ in DAG and a set $Z$ that $d$-separates $X$ from $Y$, $Z$ is a minimal separator if no proper subset of $Z$ $d$-separates $X$ from $Y$. There are polynomial time algorithms to find minimal separators[4].*

## 5.2 Benefits of Graphical Models

Although the results in this paper could have been obtained from conditional independencies alone oblivious to causal directionalities in the m-graph, it behooves us to emphasize that the power of graphical models stem from its causal interpretation. It is only through such interpretation that a user can decide the plausibility of the conditional independence assumptions embedded in the graph. The non-graphical literature on missing data, though based on conditional independencies, rarely pays attention to the cognitive question of whether researchers are capable of judging the plausibility of those assumptions. Another advantage of graphical models, be they causal or probabilistic, is that the sum total of conditional independencies that follow from a given set of assumptions are explicitly represented in the graph and need not be derived by lengthy procedures involving say, probability theory or graphoid axioms.

## 5.3 Constrained Recoverability

For any set $S \subseteq V_m$, let $R_S$ represent the set of $R$ variables corresponding to variables in $S$. In any observation $(r, v^*, v_o)$, letting $S \subseteq V_m$ be the set of observed variables, we have

$$P(r, v^*, v_o) = P(R_S = 0, R_{V_m \setminus S} = 1, s, v_o). \tag{10}$$

We assume that we are given a missingness pattern that specifies which sets of variables in $V_m$ are never observed simultaneously and which could be observed simultaneously. We will represent the *observed missingness pattern* (more precisely observability pattern) as a collection $\mathbb{C}$ of sets $S \subseteq V_m$ such that $P(R_S = 0, R_{V_m \setminus S} = 1, s, v_o) > 0$ for some $(s, v_o)$ values. We denote $\overline{\mathbb{C}}$ the collection of the rest of sets $S$ in $V_m$ with $P(R_S = 0, R_{V_m \setminus S} = 1, s, v_o) = 0$ for all $s$ and $v_o$ values.

**Definition 5** (Recoverability). *Given a m-graph $G$, and observed missingness pattern $\mathbb{C}$, a target probabilistic relation $Q$ defined on the variables in $V$ is said to be* recoverable *if there exists an algorithm that produces a consistent estimate of $Q$. In other words, $Q$ is recoverable if it can be expressed in terms of the observed strictly positive probabilities $P(R_S = 0, R_{V_m \setminus S} = 1, S, V_o)$ for $S \in \mathbb{C}$ - that is, if $Q^{M_1} = Q^{M_2}$ for every pair of models $P^{M_1}(V, U, R)$ and $P^{M_2}(V, U, R)$ compatible with $G$ with $P^{M_1}(R_S = 0, R_{V_m \setminus S} = 1, S, V_o) = P^{M_2}(R_S = 0, R_{V_m \setminus S} = 1, S, V_o) > 0$ for all $S \in \mathbb{C}$.*

Note that for recoverability it is not necessary to require $P^{M_1}(R_S = 0, R_{V_m \setminus S} = 1, S, V_o) = P^{M_2}(R_S = 0, R_{V_m \setminus S} = 1, S, V_o) = 0$ for all $S \in \overline{\mathbb{C}}$.

2. Tian, Jin, Azaria Paz, and Judea Pearl. Finding minimal d-separators. Computer Science Department, University of California, 1998.

## 5.4 Recoverable-MNAR vs Rubin's MAR: A brief discussion

Fig. 1(d) is an example of a problem which we label MNAR and which permits recoverability of $P(X, Y)$. Among others, the following conditional independence claims hold in Fig 1(d): $Y \perp\!\!\!\perp (R_x, R_y)|X$ and $X \perp\!\!\!\perp R_x$. Given data from this example conditional independence such as $P(R_x = 1, R_y = 0|X, Y) = P(R_x = 1, R_y = 0|Y)$ required by MAR is not dictated by the graph and so, it will be violated by all but a small fraction of the distributions generated by the graph. Each distribution that violates this equality would be labeled MNAR by Rubin[25] and would be considered recoverable-MNAR in our graph-based taxonomy. The same holds for all examples that we labeled MNAR. In fact, only exceptional distributions may have a chance of being classified as MAR.

## 5.5 Necessity Proof for Theorem 2

**Lemma 3.** $P(X)$ *is not recoverable in a m-graph $G$ over $(V, U, R)$ containing a single edge $X \rightarrow R_X$.*

*Proof.* To prove non-recoverability of $P(X)$ we present two models compatible with $G$:

$$P^{M_1}(v, u, r) = P_1(x, r_X) \prod_{i, V_i \neq X} P(v_i) \prod_j P(u_j) \prod_{k, R_k \neq R_X} P(r_k), \qquad (11)$$

$$P^{M_2}(v, u, r) = P_2(x, r_X) \prod_{i, V_i \neq X} P(v_i) \prod_j P(u_j) \prod_{k, R_k \neq R_X} P(r_k). \qquad (12)$$

We construct $P_1(x, r_X)$ and $P_2(x, r_X)$ as given in Table 1 such that they agree on the observed distributions: $P_1(X, R_X = 0) = P_2(X, R_X = 0) > 0$ and $P_1(R_X = 1) = P_2(R_X = 1) > 0$, but disagree on the query $P_1(X) \neq P_2(X)$.

| $X$ | $R_X$ | $P_1(X, R_X)$ | $P_2(X, R_X)$ |
|-----|-------|---------------|---------------|
| 0 | 0 | 1/3 | 1/3 |
| 1 | 0 | 1/3 | 1/3 |
| 0 | 1 | 0 | 1/3 |
| 1 | 1 | 1/3 | 0 |

Table 1: Two distributions for $X \rightarrow R_X$.

Then we have that the two models agree on all the observed distributions:

$$P^{M_i}(R_S = 0, R_X = 0, R_{V'_m \setminus S} = 1, x, s, v_o) = P_i(R_X = 0, x)P(R_S = 0, R_{V'_m \setminus S} = 1, s, v_o), \quad i = 1, 2, \tag{13}$$

and

$$P^{M_i}(R_S = 0, R_X = 1, R_{V'_m \setminus S} = 1, s, v_o) = P_i(R_X = 1)P(R_S = 0, R_{V'_m \setminus S} = 1, s, v_o), \quad i = 1, 2, \tag{14}$$

where $V'_m = V_m \setminus \{X\}$ and $S \subseteq V'_m$. But $P^{M_1}(x) = P_1(x)$ disagrees with $P^{M_2}(x) = P_2(x)$. □

**Lemma 4.** *If a target relation $Q$ is not recoverable in m-graph $G$, then $Q$ is not recoverable in the graph $G'$ resulting from adding a single edge to $G$.*[5]

*Proof.* If $Q$ is not recoverable in $G$, then there exist two models $P^{M_1}(V, U, R)$ and $P^{M_2}(V, U, R)$ compatible with $G$ decomposed as

$$P^{M_k}(v, u, r) = \prod_i P^{M_k}(v_i|pa_i^v) \prod_j P^{M_k}(u_j|pa_j^u) \prod_l P^{M_k}(r_l|pa_l^r), \quad k = 1, 2, \tag{15}$$

such that, for all $S \subseteq V_m$

$$P^{M_1}(R_S = 0, R_{V_m \setminus S} = 1, S, V_o) = P^{M_2}(R_S = 0, R_{V_m \setminus S} = 1, S, V_o) > 0, \qquad (16)$$

and

$$Q^{M_1} \neq Q^{M_2}. \qquad (17)$$

For the graph $G'$, we can specify model parameters in such a way that the extra edge added to $G$ is ineffective and hence construct the same distributions as $M_1$ and $M_2$. Without loss of generality, assuming $G'$ is obtained from $G$ by adding edge $X \rightarrow V_q$ where $X$ could be a $V$ or $U$ variable. We construct two models $M_1'$ and $M_2'$ compatible with $G'$ with parameters given by

$$P^{M_k'}(v_q | pa_q^v, x) = P^{M_k}(v_q | pa_q^v), \quad k = 1, 2, \qquad (18)$$

$$P^{M_k'}(v_i | pa_i^v) = P^{M_k}(v_i | pa_i^v), \quad i \neq q, \quad k = 1, 2, \qquad (19)$$

$$P^{M_k'}(u_j | pa_j^u) = P^{M_k}(u_j | pa_j^u), \quad \forall j, \quad k = 1, 2, \qquad (20)$$

$$P^{M_k'}(r_l | pa_l^r) = P^{M_k}(r_l | pa_l^r), \quad \forall l, \quad k = 1, 2. \qquad (21)$$

Clearly $P^{M_k'}(v, u, r) = P^{M_k}(v, u, r), k = 1, 2$. Therefore the two models $M_1'$ and $M_2'$ also satisfy Eqs. (16) and (17). And we conclude $Q$ is not recoverable in $G'$. The same arguments apply if $G'$ is obtained from $G$ by adding a parent to $U$ or $R$ variable. $\qquad \square$

## 5.6 Proof of Theorem 3

*Proof.* Since the model is Markovian, $P(v)$ may be decomposed as

$$P(v) = \prod_{i, V_i \in V_o} P(v_i | pa_i^o, pa_i^m) \prod_{j, V_j \in V_m} P(v_j | pa_j^o, pa_j^m). \qquad (22)$$

$R_{Pa_i^m}$ must be non-descendants of $V_i$, otherwise they will be descendants of $Pa_i^m$. Therefore $V_i \perp\!\!\!\perp R_{Pa_i^m} | (Pa_i^o \cup Pa_i^m)$. Similarly, $R_{V_j}$ and $R_{Pa_j^m}$ must be non-descendants of $V_j$ and we have $V_j \perp\!\!\!\perp (R_{V_j} \cup R_{Pa_j^m}) | (Pa_j^o \cup Pa_j^m)$. Using these conditional independences we obtain Eq. (9) from (22). $\qquad \square$

## 5.7 Proof of Lemma-1

*Proof.* Let the order be $O = V_1, V_2, V_3, ... V_n$. The factorization corresponding to $O$ is :

$$P(V_1, .., V_n) = \prod_j P(V_j | V_{j+1}, ..., V_n) = P(V_i | V_{i+1}, ... V_n) \prod_{j \neq i} P(V_j | V_{j+1}, ..., V_n)$$

If there is no (minimal) separator $S$ such that $S \subseteq \{V_{i+1}, ... V_n\}$ then we must have $V_i \not\perp\!\!\!\perp R_{V_i} | V_{i+1}, ... V_n$. Thus we have shown that there exists a term $P(V_i | V_{i+1}, ... V_n)$ in the factorization that does not satisfy the condition in Theorem-4, thereby making $O$ non-admissible. $\qquad \square$

## Footnotes

[3]Pearl, Judea. Causality: models, reasoning and inference, Cambridge Univ press, 2009.

[4]1. Acid, Silvia, and Luis M. De Campos. "An algorithm for finding minimum d-separating sets in belief networks." Proceedings of the Twelfth international conference on Uncertainty in artificial intelligence. Morgan Kaufmann Publishers Inc., 1996.

[5]This lemma and its proof closely follow Lemma 13 in *J. Tian and J. Pearl, On the identification of causal effects, UCLA Cognitive Systems Laboratory, Technical Report (R-290-L), 2003.*