[Reviews · NeurIPS 2013]

Submitted by Assigned_Reviewer_6

This is an interesting paper - the application of graphical methods to analyze missing data patterns may prove to be very useful.

The paper contains the word "causal graph" in the title and in the introduction. However, none of the results seem to depend on the graph being causal. The results are entirely about conditional independence and factorizations. Even if we were in possession of a true causal graph, presumably we could use any other Markov equivalent DAG and we would derive the same results (?)
Summary: An interesting and thought provoking paper.

However, the authors could do a better job of linking to existing literature.


Submitted by Assigned_Reviewer_7

Review Summary: This paper presents a graphical models approach to reasoning about when probabilities can be computed in the presence of missing data. The approach hinges on definitions of MCAR and MAR in terms of conditional independencies between data and response indicator variables. The MAR definition is considerably stronger than the standard definition studied by Little and Rubin, which encodes the minimal assumption required to obtain unbiased estimates while ignoring the missing data mechanism. The current work is certainly novel and interesting. It appears to be technically correct given the initial definitions. However, the main component the paper is missing is a discussion of the impact of changing the definitions. Detailed comments are provided below.

Novelty/Originality: The work appears to be novel. The fact that the authors are able to provide a treatment of the missing data taxonomy using graphical models is interesting and stems from their MCAR/MAR definitions based on independence between random variables. The standard definition of MAR is a statement of symmetry within the distribution P(R|X) that depends on the actual values of R and X. It is not a statement of independence between random variables, so Bayesian networks can not be used to describe it.

Technical Correctness: Given the initial MCAR/MAR/MNAR definitions, the remaining technical development appears to be correct. The authors show that they can apply their framework to reason effectively about what queries are recoverable, even in some fairly complex situations.

The main issue with the paper is the definitions themselves. Defining MAR at the random variable level means that data sets that would be considered MAR by Little and Rubin could correspond to graphs considered MNAR here. This creates some confusion with the claims in the paper.

The authors repeatedly make claims like "Though it is widely believed that relations in MNAR data sets are not recoverable, [we] demonstrate otherwise" (lines 229-230). This is confusing since the majority of work uses the standard definition of MNAR from Little and Rubin, which differ from the definitions used here. Are the authors saying that under *their* definition of MNAR, it is widely believed that queries are not recoverable?

Additionally, the authors should clarify what the relationship is between the the recoverable MNAR cases under their framework, and how data from the corresponding models would be labeled by Little and Rubin's definitions. For example, is it the case that all of the MNAR graphs presented here that admit recoverable queries would generate data that Little and Rubin would consider MAR? Are any of the examples shown graphs that are NAMR under the authors' definition, but admit recoverable queries, and would also generate data considered MNAR by Little and Rubin?

Significance/Impact: The significance of the work is difficult to judge without the clarifications requested above. There could be some interest in these alternative definitions and the example applications of the framework are quite nice. On the other hand, the limitations have not been described by the authors at all.

An additional issue with the paper is the question of where the structures that are reasoned about come from? Given a novel domain, coming up with a model over the primary variables can be challenging. The additional complexity to soliciting a joint data/response model would be quite onerous. Given possibly NMAR data, what are the prospects for learning these model structures? Given two candidate structures, one in which a given query is recoverable and another where it is not, can the framework itself provide any guidance on which structure to choose?

Presentation/Clarity: The paper is well written although quite dense. Focusing the development around specific examples does help, but the examples themselves become quite complex.

* The name of the model is introduced twice on page 3 (lines 80, 115, 135).

* In equation 1, what does the value m stand for? It is not introduced prior to this point in the paper. I assume it stands for "missing"?

* I would consider a more politically correct example on line 149.
Summary: The paper presents an interesting graphical model-based framework for reasoning about missing data. The implications of changing the traditional MCAR/MAR/MNAR definitions needs to be discussed at much greater length.

Submitted by Assigned_Reviewer_8

The paper looks at the problem of estimation in the presence of missing values. Given a dataset with missing values, a model for the missingness mechanism, and a relation Q among variables that needs to be estimated from data, the paper establishes conditions under which an unbiased estimate of Q can be recovered from the data. It also suggests a heuristic for finding an ordered set of conditional distributions which can be estimated to compute an unbiased estimate of Q. The theoretical analysis covers the relatively simpler missingness mechanisms of MCAR and MAR, as well as the more difficult case of MNAR.

This is a valuable paper because it proposes a single formulation that captures all three cases of MCAR, MAR, and MNAR, and uses it to analyze when unbiased estimation is possible. Most of the ML and Statistics literature on missing values only deals with the easier cases of MCAR and MAR and mostly ignores the more difficult MNAR case. By initiating a theoretical analysis of all three, and in particular MNAR, the paper is making a significant contribution to the missing values literature. By building on this work, it should be possible to come up with imputation algorithms that can handle even MNAR missingness. There is still some gap between the theoretical analysis presented in the paper and actually analyzing the behaviour of existing popular imputation algorithms (e.g. Multiple Imputation using Chained Equations) in Stats/ML literature for different types of missingness, but this is a valuable first step. Further work will hopefully close the gap.
Summary: The paper makes a useful first step towards building a theoretical understanding of when unbiased estimation is possible under different missingness mechanisms. It should lead to new algorithms for handling missing values and possibly better understanding of existing imputation algorithms.

Submitted by Assigned_Reviewer_9

The authors propose a framework which uses causal DAGs to model missing data problems and identify situations where an algorithm exists to derive a unbiased estimator for a specific query given the structure of the missing data. The causal DAG is used to model how the 'missingness process' interacts with the variables. The authors use this graph (and the d-separation relations) to derive sufficient conditions over the missingness process (or missingness graph) for the derivability of an unbiased estimator.

While finding unbiased estimates when data are not missing at random is an important problem which has real world applications in areas such recommendation systems, the impact of the results and framework of this paper appears to be very limited. The authors show, using their framework, when data are missing completely at random (MCAR) and missing at missing at random (MAR), i.e. the missing variables are independent of the missingness process given the observed variables, an unbiased estimator can be derived, but this is not a new result. The conditions the authors derive for other cases, where data are missing not at random (MNAR), seem to be technical results which follow, but it general will have very limited application to real world problems because they require knowledge of the missingness process that will be unknown. The authors do provide an example early on where data are MNAR and the relevant information about the missingness process is knowable (a study where (i) cases who underwent treatment X did not report outcome Y, and then (ii) a handful of treatment values are accidentally deleted), but this example seems rather contrived. If it is the case that sufficient knowledge about the missingness process can be known so that the author's MNAR conditions can be applied in commonly occuring real world examples resulting in improvements in areas such recommendation systems, then the authors did not sufficiently motivate their procedure and make the connections showing how their procedure can affect these real applications commonly in practice.

Aside from the motivational/impact issues, the theoretical framework is interesting and the technical results appear to be sound (though I did not thoroughly check proofs).
Summary: The authors present a framework using causal graphs for identifying situations where unbiased estimators can be derived when data are missing possibly not at random. The approach appears sound, but unlikely to have significant real world applications since it requires knowledge of the data missingness process that often will be unknowable.
Author Feedback

Author rebuttal: Rebuttal (Edited)

We are grateful to the reviewers for taking the time to suggest needed improvements in our paper. The suggestions made by Reviewer_6 are very reasonable and shall be incorporated in the paper. In particular, we shall refer to the literature on CAR and replace the term "causal graph" by "graphical models"(the former is necessary to make the latter meaningful).

The clarifications requested by Reviewer_7 are provided below:

1. We are indeed saying that "it is widely believed that under our definition of MNAR queries are not recoverable". It is also widely believed that queries are not recoverable under Rubin's MNAR. There are several papers and books that: (a) share these 2 beliefs, and (b) explicitly state that recoverability under both definitions of MNAR is largely unexplored.

2. No, it is not the case that all the recoverability-permitting MNAR graphs presented in our paper would generate data that Little and Rubin would consider MAR. The overwhelming majority of the data generated by each one of our examples will not be considered MAR by Little and Rubin. The following example explains why.

Fig 1(d) in our paper is an example of a problem which we label MNAR. Among others, the following conditional independence claims hold in fig 1(d): Y || Rx,Ry |X and X || Rx. From this we conclude that P(X,Y) is recoverable. Now, let us consider how data from this example are classified by Little and Rubin. The conditional independence such as P(Rx=1,Ry=0 | X,Y) = P(Rx=1,Ry=0| Y) required by Rubins MAR is not dictated by the graph and so, it will be violated by all but a small fraction of the distributions compatible with the graph. Each such distribution would be labeled MNAR by Little and Rubin and recoverable-MNAR by our taxonomy. The same holds for each and every one of our examples of recoverable MNAR. In fact, only exceptional distributions may have a chance of being classified as MAR by Little and Rubin.

A discussion on these lines clarifying the relation between recoverable MNAR and Rubin's MAR shall be included in the appendix to our paper and we thank you for bringing this potential confusion to our attention.

3. Yes, our framework can be used to distinguish between two structures, one in which a given query is recoverable and another in which it is not. Exceptions occur when the two competing models are statistically "indistinguishable" - a property we define and algorithmize in a follow-up paper (http://ftp.cs.ucla.edu/pub/stat\_ser/r415.pdf). Please note that some of our tests are powerful enough to rule out MAR and, based on the data alone, place a problem in the MNAR category.


Our main rebuttal concerns the rating of our paper as 'incremental' by reviewers 7 and 9, citing difficulties in learning or verifying the structures of the models analyzed. We request the reviewers to kindly re-weigh their judgment in light of the following considerations:

Although we cannot always be sure of the structure of the model, understanding what model features make algorithms successful or unsuccessful is essential for devising algorithms that cover large sets of possible scenarios. Rubin's seminal work is an example of such analysis. It would not have been initiated had he been inhibited by the impossibility of ascertaining (from data alone) whether a given data set is truly MAR.

Understanding what the world must be like for one's procedure to work is an important component in designing and improving our algorithms. For example, if an algorithm fails to perform as expected, it is important for the user to know whether: (1) it is a theoretical impediment (non-recoverability) that accounts for the failure or (2) a mismatch between the algorithm and the data-generation mechanism (e.g., using multiple imputation on an MNAR problem might cause such mismatch). Our procedures can generate tentative warning signals in each case.

Although our paper focuses on the deductive approach - going from a hypothesized model to its consequences, it does not mean that the work is "incremental" and "unlikely to have much impact". Giving rank-and-file researchers and users the tools to take any hypothesized model and determine (a) whether a query is recoverable and (b) whether the model has testable implications will, in our opinion, have a major and lasting impact on the way researchers will approach missing data problems in the future. It will!!!

We would greatly appreciate if you would kindly take these additional impacts into consideration while making your decision.